The systematic position of the enigmatic thyreophoran dinosaur Paranthodon africanus, and the use of basal exemplifiers in phylogenetic analysis

Raven Thomas J. tom.raven13@imperial.ac.uk 1 2 3
Maidment Susannah C.R. 2 3
1 Department of Earth Science & Engineering, Imperial College London , London , United Kingdom
2 School of Environment & Technology, University of Brighton , Brighton , United Kingdom
3 Department of Earth Sciences, Natural History Museum , London , UK
Farke Andrew
Electronic publication date: 2018 Mar 20
Publication date: 2018
Volume: 6
Electronic Location ID: e4529
Received 2018 Jan 8; Accepted 2018 Mar 2
Copyright: ©2018 Raven and Maidment
Copyright year: 2018
Copyright holder: Raven and Maidment
License: This is an open access article distributed under the terms of the Creative Commons Attribution License, which permits unrestricted use, distribution, reproduction and adaptation in any medium and for any purpose provided that it is properly attributed. For attribution, the original author(s), title, publication source (PeerJ) and either DOI or URL of the article must be cited.
License URL: https://creativecommons.org/licenses/by/4.0/

Keywords: Systematics, Thyreophora, Phylogenetics, Exemplifiers

Funding: The authors received no funding for this work.

==============================
The first African dinosaur to be discovered, Paranthodon africanus was found in 1845 in the Lower Cretaceous of South Africa. Taxonomically assigned to numerous groups since discovery, in 1981 it was described as a stegosaur, a group of armoured ornithischian dinosaurs characterised by bizarre plates and spines extending from the neck to the tail. This assignment has been subsequently accepted. The type material consists of a premaxilla, maxilla, a nasal, and a vertebra, and contains no synapomorphies of Stegosauria. Several features of the maxilla and dentition are reminiscent of Ankylosauria, the sister-taxon to Stegosauria, and the premaxilla appears superficially similar to that of some ornithopods. The vertebral material has never been described, and since the last description of the specimen, there have been numerous discoveries of thyreophoran material potentially pertinent to establishing the taxonomic assignment of the specimen. An investigation of the taxonomic and systematic position of Paranthodon is therefore warranted. This study provides a detailed re-description, including the first description of the vertebra. Numerous phylogenetic analyses demonstrate that the systematic position of Paranthodon is highly labile and subject to change depending on which exemplifier for the clade Stegosauria is used. The results indicate that the use of a basal exemplifier may not result in the correct phylogenetic position of a taxon being recovered if the taxon displays character states more derived than those of the basal exemplifier, and we recommend the use, minimally, of one basal and one derived exemplifier per clade. Paranthodon is most robustly recovered as a stegosaur in our analyses, meaning it is one of the youngest and southernmost stegosaurs.

Introduction

The first dinosaur to be found in Africa, Paranthodon africanus (NHMUK [Natural History Museum, London, UK] R47338), was discovered in 1845 in the Kirkwood Formation of South Africa. Originally identified as the pareiasaur Anthodon serranius (Owen, 1876), then the ankylosaurian Palaeoscincus africanus (Broom, 1910) and then the stegosaurian Paranthodon oweni (Nopsca, 1929), the specimen has had uncertain taxonomical affinities. Finally, Galton & Coombs (1981) settled the nomenclatural debate and coined Paranthodon africanus, agreeing with the assignment to Stegosauria. Stegosauria is a clade of thyreophoran ‘armoured’ ornithischian dinosaurs, characterized by the possession of two bizarre parasagittal rows of plates and spines that extend from the head to the end of their tail. They have a restricted temporal range, from the Middle Jurassic to the Lower Cretaceous, and are known from strata worldwide, with particularly high biodiversity in the Middle and Upper Jurassic of China (Maidment et al., 2008).

Dating the Kirkwood Formation, where Paranthodon was discovered, has proven problematic. However, recent consensus suggests the fossiliferous sections of the Upper Kirkwood Formation date to the early Early Cretaceous (e.g., Forster et al., 2009; Choiniere, Forster & De Klerk, 2012; McPhee et al., 2016). This would make Paranthodon one of the youngest stegosaurs (Pereda Suberbiola et al., 2003), and stratigraphically close to the assumed extinction of the group. The Kirkwood Formation is part of the Uitenhage Group, found within the Algoa Basin of South Africa (Muir, Bordy & Prevec, 2015), and consists of three members; the Swartkops Member, the Colchester Member and an unnamed stratigraphically higher unit, which contains all of the vertebrate fossil material found in the Kirkwood Formation (McPhee et al., 2016). The lithologic description of the upper unit by McPhee et al. (2016) matches the matrix of NHMUK R47338, in that it is an olive-grey medium sandstone, and thus it is likely that Paranthodon is derived from this unit. The geographic location of Paranthodon is particularly significant because it represents one of only two Gondwanan stegosaurs (Mateus, Maidment & Christiansen, 2009), although Han et al. (2017) also found the Argentinian dinosaur Isaberrysaura to be a stegosaur.

The first phylogeny focusing on Stegosauria was produced by Galton & Upchurch (2004), but this provided little resolution in the morphologically conservative clade, and Paranthodon was deleted a posteriori from the analysis in order to achieve higher resolution. Maidment et al. (2008) (later updated for new taxa in Mateus, Maidment & Christiansen (2009); Maidment (2010)) was the first phylogenetic analysis to include Paranthodon, but found it in a polytomy towards the base of Stegosaurinae with Loricatosaurus priscus and Tuojiangosaurus multispinus. The most recent phylogeny of Stegosauria by Raven & Maidment (2017) found Paranthodon in a sister-taxon relationship with Tuojiangosaurus, which together were sister-taxa to the clade Huayangosauridae (Huayangosaurus taibaii + Chungkingosaurus jiangbeiensis).

The material assigned to Paranthodon is a left partial maxilla, premaxilla and nasal (Fig. 1), and two referred teeth (Maidment et al., 2008). Additionally, there is a partial vertebra that was not described by Galton & Coombs (1981). Although classified as a stegosaurian, there are features that are reminiscent of the Ankylosauria, the sister clade to Stegosauria. These include tooth morphology and the presence of a secondary maxillary palate (Vickaryous, Maryańska & Weishampel, 2004). Furthermore, the dorsally elongate premaxilla is dissimilar to that of other thyreophorans (Galton & Upchurch, 2004). This study provides a detailed re-description of the material referred to Paranthodon, including previously undescribed material, and provides comprehensive anatomical comparisons in order to evaluate the systematic position of the taxon. Furthermore, this study utilises numerous phylogenetic hypotheses to constrain the evolutionary relationships of Paranthodon, including the first analysis of the taxon in an ankylosaurian phylogeny.

Figure 1 Comparison of cranial material of Paranthodon africanus NHMUK R47338 with that of Stegosaurus.

Grey section, material of Paranthodon, including partial premaxilla, maxilla and nasal. Stegosaurus skull is a reconstruction from Stegosaurus stenops USNM 4934 (United States National Museum) and DMNH 2818 (Denver Museum of Nature and Science).

Systematic Palaeontology

DINOSAURIA Owen, 1841	
ORNITHISCHIA Seeley, 1887	
THYREOPHORA Nopcsa, 1915 (sensu Norman, 1984)	
STEGOSAURIA Marsh, 1877	
Paranthodon Nopcsa, 1929	
Paranthodon africanus Broom, 1910	

Synonymy

Anthodon serrarius Owen, 1876	
Palaeoscincus africanus Broom, 1910	
Paranthodon oweni Nopcsa, 1929	

Holotype: NHMUK R47338. Left partial maxilla, premaxilla, nasal and a dorsal vertebra.

Previously referred specimen: NHMUK R4992. Two teeth. Locality and horizon unknown. Maidment et al. (2008) noted that while the teeth appear similar in morphology to Paranthodon, there are no autapomorphies of the genus located on the teeth, and so they were regarded as indeterminate stegosaurian. However, as there are no synapomorphies of Stegosauria located on the teeth, they are referred to as indeterminate thyreophoran herein.

Diagnosis: The only identifiable autapomorphy of this genus within Stegosauria is the possession of a medially extending maxillary palate.

Occurrence: Bushmans River, Algoa Basin, Eastern Cape Province, South Africa. Upper Kirkwood Formation, early Early Cretaceous (possibly Berriasian-Valanginian, Choiniere, Forster & De Klerk, 2012; McPhee et al., 2016).

Remarks: The placement of Paranthodon within Stegosauria herein is based on morphological similarities with stegosaurs, as well as numerous phylogenetic analyses in this study (see ‘Discussion’ for further information). In stegosaurian, ankylosaurian and basal ornithischian cladograms, Paranthodon is found within Stegosauria or sister-taxon to the stegosaurian exemplifier used. Although Paranthodon contains no synapomorphies that place it unequivocally in Stegosauria, the use of phylogenetics allows this referral, and therefore Paranthodon can be considered a valid genus due to the presence of an autapomorphy within Stegosauria.

Description

The last description of Paranthodon (NHMUK R47338) was by Galton & Coombs (1981), but the discovery of new thyreophoran material means a re-description is warranted. The previous study misidentified part of the posterior process of the premaxilla as the nasal, and there was no description of the vertebra, which is described here for the first time. Measurements are found in Table 1.

Table 1 Measurements of the elements of Paranthodon africanus NHMUK R47338 and previously referred specimen NHMUK R4992.

Measurement	Specimen	
	NHMUK R47338	NHMUK R4992	
Nasal, anteroposterior length	134 mm		
Nasal, width	63 mm		
Nasal, dorsoventral height	33 mm		
Premaxilla and maxilla, anteroposterior length	178 mm		
Premaxilla and maxilla, width	67 mm		
Premaxilla and maxilla, dorsoventral height	82 mm		
Mean tooth crown height	3.04 mm	4.25 mm	
Mean tooth crown anteroposterior length	6.20 mm	5.25 mm	
Mean tooth cingula height	2.92 mm	1.75 mm	
Mean tooth cingula anteroposterior length	7.52 mm	7.50 mm	
Mean tooth crown width	1.89 mm	1.25 mm	
Mean tooth cingula width	5.05 mm	4.25 mm	

Premaxilla

The left premaxilla consists of an anteriorly-projecting anterior process and a posterior process that projects posterodorsally (Fig. 2). The anterior end of the premaxilla is incomplete, but the anterior process is sinuous in lateral view and curves ventrally, as in the stegosaurs Miragaia (Mateus, Maidment & Christiansen, 2009) and Huayangosaurus (Sereno & Dong, 1992), the ankylosaur Silvisaurus (NHMUK R1107) and the basal ornithischian Heterodontosaurus (Butler, Porro & Norman, 2008). This, however, contrasts to the horizontally-projecting process of the stegosaurs Chungkingosaurus (Maidment & Wei, 2006) and Stegosaurus stenops (NHMUK R36730), the ankylosaur Edmontonia (NHMUK R36851), and the basal ornithischian Lesothosaurus (Sereno, 1991). The posterior process of the premaxilla is robust and similar to that of the basal ornithischian Heterodontosaurus (Butler, Upchurch & Norman, 2008) and the ornithopods Camptosaurus (NHMUK R1608) and Jinzhousaurus (Wang & Xu, 2001) in that it intervenes between the maxilla and nasal to stop them contacting each other. The angle of the posterior process in Paranthodon is 47 degrees relative to horizontal, although this varies widely in thyreophorans (Table 2). The premaxilla is edentulous, as in every other stegosaur with cranial material preserved other than Huayangosaurus (Sereno & Dong, 1992). The distribution of premaxillary teeth in other ornithischians varies; basal members of most ornithischian groups possess premaxillary teeth. For example, the basal ornithopod Hypsilophodon has five (Norman et al., 2004), and basal ankylosaurs, such as Gargoyleosaurus, Pawpawsaurus and Cedarpelta (Kinneer, Carpenter & Shaw, 2016) possess premaxillary teeth. More derived members of Ornithopoda and Ankylosauria, however, have edentulous premaxillae (e.g., most basal iguanodontids (Norman et al., 2004); Edmontonia (NHMUK R36851); Anodontosaurus (NHMUK R4947)). The premaxillae contacted each other along a dorsoventrally deep sutural surface , and this forms a small premaxillary palate, similar to that of Stegosaurus stenops (NHMUK R36730) and in the ankylosaur Gastonia (Kinneer, Carpenter & Shaw, 2016), but not as robust as that of the basal thyreophoran Scelidosaurus (NHMUK R1111). The premaxillary palate of Paranthodon has a transversely concave dorsal surface. Despite poor preservation, the external naris appears to face anterolaterally, as in the ankylosaurs Gastonia (Kinneer, Carpenter & Shaw, 2016) and Anodontosaurus (NHMUK R4947) and the ornithopods Camptosaurus (NHMUK R1608) and Jinzhousaurus (Wang & Xu, 2001). This feature is, however, variable in stegosaurs; the same condition is seen in Huayangosaurus (Sereno & Dong, 1992), yet in Stegosaurus (NHMUK R36730) and Hesperosaurus (Carpenter, Miles & Cloward, 2001), the external nares face anteriorly. The external naris is longer anteroposteriorly than wide transversely in Paranthodon, similar to other stegosaurs such as Stegosaurus stenops (NHMUK R36730) and Chungkingosaurus (Maidment & Wei, 2006), and ornithopods such as Camptosaurus (NHMUK R1608) and Hypsilophodon (NHMUK R197). The condition is the same in the ankylosaurs Silvisaurus (NHMUK R1107), Europelta (Kirkland et al., 2013) and Kunbarrasaurus (Leahey et al., 2015); in contrast, in the ankylosaurs Anodontosaurus (NHMUK R4947) and Edmontonia (NHMUK R36851) the naris is wider transversely than it is long anteroposteriorly. The internal surface of the naris is smooth, as in Europelta (Kirkland et al., 2013); this suggests the narial passage was simple, rather than convoluted as in ankylosaurids and derived nodosaurids (Witmer & Ridgely, 2008).

Figure 2 Premaxilla and maxilla of Paranthodon africanus NHMUK R47338.

(A) Medial; (B) lateral; (C) posterior; (D) dorsal; (E) ventral; (F) anterior views. pmp, premaxillary process; smp, secondary maxillary process; pp, posterior process; ap, anterior process. Images copyright The Natural History Museum.

Table 2 Premaxillary posterior process angle across a range of ornithischians.

Taxon	Premaxilla posterior process angle, relative to horizontal (°)	
Camptosaurus dispar	40	
Gastonia burgei	60	
Hesperosaurus mjosi	40	
Heterodontosaurus tucki	40	
Huayangosaurus taibaii	30	
Hypsilophodon foxii	75	
Jinzhousaurus yangi	60	
Paranthodon africanus	47	
Scelidosaurus harrisonii	60	
Stegosaurus stenops	16	
Tenontosaurus tilletii	50	

Maxilla

The maxilla is triangular in lateral view, with the tooth row forming an elongate base of the triangle (Fig. 2). This is similar to the condition in most other thyreophorans (e.g., Stegosaurus (NHMUK R36730), Hesperosaurus (Carpenter, Miles & Cloward, 2001), Silvisaurus (NHMUK R1107) and Edmontonia (NHMUK R36851)). However, the maxilla of the basal ankylosaur Kunbarrasaurus is rectangular with the long axis orientated dorsoventrally (Leahey et al., 2015), and the element is rectangular in the ornithopods Camptosaurus (NHMUK R1608) and Jinzhousaurus (Wang & Xu, 2001), with the long axis anteroposterior. In lateral view, the maxillary tooth row is horizontal, as in the ornithopod Camptosaurus (NHMUK R1608), and the stegosaurs Stegosaurus (NHMUK R36730) and Huayangosaurus (Sereno & Dong, 1992). This contrasts with many ankylosaurs, such as Silvisaurus (NHMUK R1107), Europelta (Kirkland et al., 2013) and Kunbarrasaurus (Leahey et al., 2015), as well as the stegosaur Hesperosaurus (Carpenter, Miles & Cloward, 2001), where the tooth row arches ventrally. In ventral view, the tooth row is not inset from the lateral edge of the maxilla and is in line with the lateral edge of the premaxilla. This is similar to the condition in the stegosaur Tuojiangosaurus (Maidment & Wei, 2006) and the basal ornithischian Lesothosaurus (Sereno, 1991), but contrasts with all other members of Thyreophora, as well as ornithopods including Hypsilophodon (NHMUK R197), where there is a laterally-extending ridge dorsal to the tooth row. The tooth row is sinuous in ventral view, as in the basal thyreophoran Scelidosaurus (NHMUK R1111), the stegosaur Jiangjunosaurus (Jia et al., 2007) and the ankylosaurs Anodontosaurus (NHMUK R4947), Gastonia (Kinneer, Carpenter & Shaw, 2016), Edmontonia (NHMUK R36851), Pawpawsaurus (Kinneer, Carpenter & Shaw, 2016), Panoplosaurus (Kirkland et al., 2013) and Silvisaurus (NHMUK R1107). In Stegosaurus (NHMUK R36730) and Huayangosaurus (Sereno & Dong, 1992) the tooth row is straight in ventral view. There is a horizontal diastema between the maxillary teeth and the maxilla-premaxilla suture, similar to that of Stegosaurus (NHMUK R36730) and the ankylosaur Silvisaurus (NHMUK R1107). This is in the same location as the oval depression seen in the stegosaur Huayangosaurus (Sereno & Dong, 1992). The contact angle between the maxilla and premaxilla in dorsal view is 30 degrees, similar to that of the stegosaurs Tuojiangosaurus (Maidment & Wei, 2006) and Huayangosaurus (Sereno & Dong, 1992). The ankylosaurs Ankylosaurus (Kinneer, Carpenter & Shaw, 2016) and Pinacosaurus (Maryańska, 1977) have a contact with no deflection along the midline. The contact is perpendicular in ornithopods such as Hypsilophodon (NHMUK R197) and Camptosaurus (NHMUK R1608). Contra Galton & Coombs (1981), who said the posterior process of the premaxilla underlaps the maxilla, the posterior process of the premaxilla overlaps the maxilla, as in the stegosaur Huayangosaurus (Sereno & Dong, 1992). The posterior portion of the maxilla is incomplete, and so there is no evidence of contact with the lacrimal or the jugal.

In medial view, the maxilla bears a ridge extending from the premaxillary palate to form a secondary maxillary palate. This feature is unknown in other stegosaurs and was considered the only identifiable autapomorphy of the genus by Maidment et al. (2008). However, it is common in ankylosaurs, including in Edmontonia (NHMUK R36851), Anodontosaurus (NHMUK R4947) and Gastonia (Kinneer, Carpenter & Shaw, 2016), although it is more pronounced than in Paranthodon. The basal thyreophorans Scelidosaurus (NHMUK R1111) and Emausaurus (Maidment, 2010) do not possess this feature.

Nasal

Only the anterior part of the left nasal is preserved (Fig. 3). It is an anteroposteriorly elongate element, as in the stegosaurs Stegosaurus (NHMUK R36730), Hesperosaurus (Carpenter, Miles & Cloward, 2001) and Huayangosaurus (Sereno & Dong, 1992), and the basal thyreophoran Scelidosaurus (NHMUK R1111). In the ankylosaur Europelta the nasal is more equidimensional (Kirkland et al., 2013), in the stegosaur Tuojiangosaurus it is triangular in dorsal view (Maidment & Wei, 2006) and in the ornithopod Jinzhousaurus it tapers anteriorly (Wang & Xu, 2001). In Paranthodon the nasal is dorsally convex, to a greater degree than in the basal thyreophoran Scelidosaurus (NHMUK R1111) but not as much as in the stegosaurs Stegosaurus (NHMUK R36730) and Hesperosaurus (Carpenter, Miles & Cloward, 2001). In the stegosaur Miragaia, this curvature is also seen, but the degree of curvature could have been affected by post-mortem deformation (Mateus, Maidment & Christiansen, 2009). In the stegosaur Tuojiangosaurus, the nasal is gently concave transversely (Maidment & Wei, 2006), as it is in the basal ornithischian Heterodontosaurus (Butler, Porro & Norman, 2008). The nasal of Paranthodon has variable dorsoventral thickness, from 2 mm to 7 mm. There are two subtle anteroposteriorly extending ridges on the dorsal surface, and it is possible these indicate the suture with the frontals, as in the stegosaur Hesperosaurus (Carpenter, Miles & Cloward, 2001). As in the basal ornithischian Heterodontosaurus, the lateral margins are thickened into nasal ridges (Butler, Porro & Norman, 2008). There is a straight suture along the midline of the nasal that would have contacted its counterpart. This is a similar depth to that of Stegosaurus (NHMUK R36730) and Hesperosaurus (Carpenter, Miles & Cloward, 2001). In the basal thyreophoran Scelidosaurus (NHMUK R1111) the sutures are not obvious and in the stegosaur Tuojiangosaurus the nasals are fused together (Maidment & Wei, 2006), although the fusion of skull sutures is likely ontogenetic in nature (Currie, Langston & Tanke, 2008). The nasal is not seen in contact with the premaxilla or maxilla, contra Galton & Coombs (1981; Fig. 1a), and is preserved separately.

Figure 3 Nasal of Paranthodon africanus NHMUK R47338.

(A) Dorsal; (B) posterior; (C) lateral; (D) ventral; (E) anterior; (F) medial. Images copyright The Natural History Museum.

Maxillary teeth

There are 13 maxillary teeth preserved, although they extend to the incomplete posterior end of the maxilla and it is possible in life the animal had more. The number of maxillary teeth among ornithischians is widely variable, ranging from 10 in the ornithopod Camptosaurus (NHMUK R1608) to as many as 35 in Ankylosaurus (Kinneer, Carpenter & Shaw, 2016); tooth count also varies intraspecifically and was likely ontogenetically controlled (Butler, Porro & Norman, 2008). There are three teeth on the medial surface of the maxilla that are erupting, and the second tooth from the maxillary diastema is not fully erupted. The teeth of Paranthodon are symmetrical with a centrally located apex, as in the stegosaurs Stegosaurus (NHMUK R36730), Miragaia (Mateus, Maidment & Christiansen, 2009), Hesperosaurus (Carpenter, Miles & Cloward, 2001), Tuojiangosaurus (Maidment & Wei, 2006), and Jiangjunosaurus (Jia et al., 2007) and the ankylosaur Gastonia (Kinneer, Carpenter & Shaw, 2016). The stegosaur Chungkingosaurus has a sharp, asymmetric tooth crown (Maidment & Wei, 2006) whereas the basal thyreophoran Scelidosaurus (NHMUK R1111) has distally offset crowns. The maxillary teeth of heterodontosaurids are chisel-shaped, with denticles restricted to the apical third of the crown (Norman et al., 2004), and in hadrosaurids they are arranged into a compact dental battery with elongate tooth crowns (Horner, Weishampel & Forster, 2004). A prominent ring-like cingulum is present on lingual and buccal sides of the teeth. This is the same in all other stegosaurs in which the teeth are known (e.g., Stegosaurus (NHMUK R36730), Tuojiangosaurus (Maidment & Wei, 2006), Hesperosaurus (Carpenter, Miles & Cloward, 2001), Jiangjunosaurus (Jia et al., 2007), Miragaia (Mateus, Maidment & Christiansen, 2009)) except Huayangosaurus, where a reduced swelling is present but not as a ring (Sereno & Dong, 1992), and Kentrosaurus where the cingulum is restricted to one side (Galton, 1988). Within Ankylosauria, most ankylosaurs, including Edmontonia (NHMUK R36851), Silvisaurus (NHMUK R1107) and Kunbarrasaurus (Leahey et al., 2015) have a prominent cingulum, but it is not seen in Gastonia (Kinneer, Carpenter & Shaw, 2016). The cingulum of the basal thyreophoran Scelidosaurus (NHMUK R1111) is weak. The cingulum of Paranthodon varies in dorsoventral thickness along the width of each tooth in the tooth row. The best-preserved tooth is the sixth from the maxillary diastema, and is in the process of erupting. There are six denticles on the mesial side of the lingual surface, and this is seen on both the distal and mesial sides of all maxillary teeth, contra Galton & Coombs (1981). The denticles curve away from the central apex and thicken towards the tooth margins. The tooth crowns of Paranthodon bear striations, extending to the cingulum, and these are confluent with the marginal denticles. The only other occurrence of this within Stegosauria is in Tuojiangosaurus (Maidment & Wei, 2006); in contrast, it is very common in ankylosaur teeth (e.g., Edmontonia (NHMUK R36851), Silvisaurus (NHMUK R1107), Gastonia (Kinneer, Carpenter & Shaw, 2016), Anodontosaurus (NHMUK R4947)). Stegosaurus (NHMUK R36730) and Kentrosaurus (Galton, 1988) have striations that extend to the cingulum, but these are not confluent with marginal denticles. The tooth root is parallel-sided, as in the stegosaur Hesperosaurus (Carpenter, Miles & Cloward, 2001), whereas the root of Kentrosaurus tapers to a point (Galton, 1988).

Vertebra

The vertebra is extremely fragmentary; only the right transverse process and prezygapophysis are identifiable (Fig. 4). The anterior edge of the prezygapophysis is broken off and so the intraprezygapophyseal shelf is not preserved. The left transverse process is not present, nor are the posterior end of the vertebra or the centrum. The top of the right transverse process is not preserved, and part of the midline ridge has split so that it tapers to a 3 mm thick slice anteriorly. The vertebra is tentatively identified as mid-dorsal based on the angle of the transverse process and the orientation of the prezygapophysis. The transverse process is elevated dorsolaterally at an angle of 60 degrees, similar to the mid-dorsal vertebrae of the stegosaurs Stegosaurus (NHMUK R36730) and Chungkingosaurus (Maidment & Wei, 2006). The dorsal vertebrae of the stegosaur Gigantspinosaurus (Maidment & Wei, 2006) have transverse processes that project laterally, whereas they project dorsolaterally in the ankylosaurs Ankylosaurus (Carpenter, 2004; Kinneer, Carpenter & Shaw, 2016), Euoplocephalus (Arbour & Currie, 2013) and Zhanghenglong (Xing et al., 2014). The transverse processes of the posterior and mid-dorsal vertebrae of Lesothosaurus are laterally orientated (Baron, Norman & Barrett, 2017), whereas on anterior dorsal vertebrae they project dorsolaterally; this shift to higher angles anteriorly is also seen in Hypsilophodon (NHMUK R197) and Heterodontosaurus (Santa Luca, 1980). In Stegosaurus (NHMUK R36730) the transverse processes are sub-horizontal in the anterior and posterior dorsal vertebrae but steeply angled in the mid-dorsal vertebrae. The parapophysis is located anteroventral to the base of the transverse process, as in the basal ornithischian Lesothosaurus (Baron, Norman & Barrett, 2017), and the stegosaur Kentrosaurus (NHMUK R16874), and is adjacent to the prezygapophysis, as in Stegosaurus sp. (NHMUK R3216). The parapophysis is more concave than Kentrosaurus (NHMUK R16874) or Stegosaurus (NHMUK R36730; NHMUK R3216). The prezygapophysis faces dorsally in Paranthodon, as in the basal ornithischian Lesothosaurus (Baron, Norman & Barrett, 2017) and the stegosaur Stegosaurus (NHMUK R36730). In contrast, the prezygapophyses of other stegosaurs face dorsomedially (Maidment, Brassey & Barrett, 2015) similarly to the condition observed in the basal ornithischian Heterodontosaurus (Santa Luca, 1980), the ornithopod Tenontosaurus (Sues & Norman, 1990), the hadrosauroid Zhanghenglong (Xing et al., 2014) and the ankylosaurs Ankylosaurus (Carpenter, 2004; Kinneer, Carpenter & Shaw, 2016) and Euoplocephalus (Arbour & Currie, 2013).

Figure 4 Vertebra of Paranthodon africanus NHMUK R47338.

(A) Anterior; (B) posterior; (C) left lateral; (D) right lateral; (E) dorsal; (F) comparison with dorsal vertebra five of NHMUK R36730 showing location of fragmentary vertebra of Paranthodon. ns, neural spine; przyg, prezygapophysis. Scale bar on left is for (A), (B), (C), (D), and (E). Scale bar on right applies to (F) only. Images copyright The Natural History Museum.

Referred teeth

There are two isolated teeth (Fig. 5) that are the previously referred specimen NHMUK R4992 (Galton & Coombs, 1981). These differ from the maxillary teeth of the holotype in that they have four denticles on either side of the slightly asymmetrical apex. The cingula are 20% of the height of the crowns, which is less than the teeth of the holotype (58–80%), although the width of the teeth is 44% of the width of the cingula, which is similar to the maxillary teeth. Similarly to the maxillary teeth, the denticles are confluent with striations that extend to the cingula. CT-scanning shows no evidence of wear facets. Details on CT-scanning methodology can be found in the Supplementary Material.

Figure 5 Previously referred teeth of Paranthodon africanus NHMUK R4992.

(A) Posterior; (B) lingual; (C) buccal; (D) anterior; (E) ventral; (F) dorsal; (G) screenshot of digital model derived from a CT-scan of one of the referred teeth, with uncertain material above crack in red. Images copyright The Natural History Museum.

Galton & Coombs (1981) hypothesised that the two teeth were from the dentary, and, more specifically, one from the left dentary. They are possibly from the dentary, due to a slight difference in morphology to the maxillary teeth; however, as the only autapomorphy of Paranthodon is on the maxilla, they cannot be referred to this genus and thus are regarded as belonging to an indeterminate thyreophoran.

Phylogenetic Methodology

Multiple phylogenetic analyses were performed to examine the phylogenetic affinities of Paranthodon.

The ankylosaurid phylogeny of Arbour & Currie (2016), the ankylosaurian phylogenies of Arbour, Zanno & Gates (2016) and Thompson et al. (2012) and the basal ornithischian phylogenies of Boyd (2015) and Baron, Norman & Barrett (2017) were updated to include Paranthodon as an Operational Taxonomic Unit (OTU) (Fig. 6). The most recent phylogeny of Stegosauria by Raven & Maidment (2017) was updated with new characters and character-scores based on a more thorough description of Paranthodon (Supplementary Data). These phylogenies were chosen as there is not currently a species-level matrix for the entirety of Thyreophora, and creating one is outside the scope of this project. All analyses were carried out in TNT (Goloboff, Farris & Nixon, 2008). The analyses were first performed on the original data matrices, using the original search settings and without including Paranthodon as an OTU, to make sure the original tree topologies could be replicated. The updated analyses were then performed using a ‘New Technology’ search, with Sect Search, Ratchet, Drift and Tree Fusing algorithms, and 10 random addition sequences. ‘Traditional’ TBR Branch-Swapping was then performed on trees held in RAM, as this provides a more complete exploration of tree space. Taxonomic exemplifiers were varied to investigate the effect on tree topology; this was done by physically eliminating taxa from the character-taxon matrix, rather than making them inactive in TNT, as deactivating taxa does not reduce the size of the grid used for the initial phase of optimisation (Goloboff & Catalano, 2016). Constraint trees were then written using the ‘Force’ command in TNT to explore how labile the position of Paranthodon was in each phylogenetic analysis. The significance of the constraint trees was tested using 1,000 replications of the Templeton Test (Salgado et al., 2017). Support for groupings was tested using symmetric resampling, which was carried out with a probability of 33% and 1,000 replicates on a ‘New Technology’ search of existing trees.

Figure 6 Simplified phylogenies from original datasets used in this study.

Ankylosaurian phylogenies by (A) Thompson et al. (2012) and (B) Arbour, Zanno & Gates (2016); (C) basal ornithischian phylogeny by Boyd (2015); (D) basal ornithischian phylogeny by Baron, Norman & Barrett (2017); (E) stegosaurian phylogeny by Raven & Maidment (2017); (F) ankylosaurid phylogeny by Arbour & Currie (2016).

Arbour & Currie (2016)

In all analyses of Arbour & Currie (2016) Lesothosaurus diagnosticus was used as the outgroup. All characters were unordered and of equal weight. The original analysis performed safe taxonomic reduction using TAXEQ3 (Wilkinson, 2001) to remove the taxa Bissektipelta archibaldi, Minmi paravertebra and Tianchisaurus nedegoapeferima, and so these taxa were also removed from all analyses here. The original analysis was repeated here, using the basal stegosaur Huayangosaurus as the exemplifier for Stegosauria, to ensure the original topology could be replicated (Analysis A1). The original analysis of Arbour & Currie (2016) used a ‘Traditional’ search, however, more common recent approaches used ‘New Technology’ searches in TNT (see Ezcurra (2016); Baron, Norman & Barrett (2017); Raven & Maidment (2017)). To test the effect of this, the original dataset was re-run with a ‘New Technology’ search with settings as previously mentioned (Analysis A2).

In Analysis A3, Paranthodon was added as an OTU, and Huayangosaurus was kept as the stegosaurian exemplifier, as in the original analysis. In Analysis A4, Paranthodon was again included as an OTU, but Huayangosaurus was replaced as the stegosaurian exemplifier by the more derived Stegosaurus. Analysis A5 included Paranthodon, Huayangosaurus and Stegosaurus as Operational Taxonomic Units.

In Analysis A6, Paranthodon was constrained to fall within Ankylosauria due to the anatomical similarities between Paranthodon and ankylosaurs. A full list of analyses and taxa used can be seen in Table 3.

Arbour, Zanno & Gates (2016)

The Arbour, Zanno & Gates (2016) dataset is essentially the same as that of Arbour & Currie (2016) but with increased taxon sampling in Nodosauridae. In all analyses, Lesothosaurus diagnosticus was used as the outgroup and all characters were unordered and of equal weight. The original analytical settings were repeated here, in order to repeat the original results (Analysis B1). As with the original analysis of Arbour & Currie (2016), a ‘Traditional’ search was used, with 1,000 random addition sequences holding 10 trees per replicate. The unedited dataset was then re-run with the more common ‘New Technology’ search (Analysis B2).

Paranthodon was then added as an OTU to the dataset, with Huayangosaurus acting as the stegosaurian exemplifier (Analysis B3). In Analysis B4, Paranthodon was again included as an OTU, but Huayangosaurus was replaced as the stegosaurian exemplifier by the more derived Stegosaurus. In Analysis B5, as well as Paranthodon and Huayangosaurus, Stegosaurus was included as an OTU. Paranthodon was then constrained to fall within Ankylosauria (Analysis B6).

Table 3 All analyses performed, including original dataset and changes applied to each iteration.

Analysis	Source of original	Settings	
Analysis A1	Arbour & Currie (2016)	Lesothosaurus used as outgroup. All characters unordered and of equal weight. Bissektipelta, Minmi paravertebra and Tianchisaurus removed. Huayangosaurus used as exemplifier for Stegosauria. ‘Traditional’ search performed with original settings of Arbour & Currie (2016).	
Analysis A2	Arbour & Currie (2016)	Same as Analysis A1, except a ‘New Technology’ search was performed.	
Analysis A3	Arbour & Currie (2016)	Same as Analysis A2, except Paranthodon was added as an Operational Taxonomic Unit.	
Analysis A4	Arbour & Currie (2016)	Same as Analysis A2, except Paranthodon and Stegosaurus were added as OTUs, and Huayangosaurus removed.	
Analysis A5	Arbour & Currie (2016)	Same as Analysis A2, except Paranthodon and Stegosaurus were added as OTUs, in addition to Huayangosaurus.	
Analysis A6	Arbour & Currie (2016)	Same as Analysis A5, except Paranthodon was constrained to fall within Ankylosauria.	
Analysis B1	Arbour, Zanno & Gates (2016)	Lesothosaurus used as outgroup. All characters unordered and of equal weight. Huayangosaurus used as exemplifier for Stegosauria. ‘Traditional’ search performed with original settings of Arbour, Zanno & Gates (2016).	
Analysis B2	Arbour, Zanno & Gates (2016)	Same as Analysis B1, except a ‘New Technology’ search was performed.	
Analysis B3	Arbour, Zanno & Gates (2016)	Same as Analysis B2, except Paranthodon was added as an Operational Taxonomic Unit.	
Analysis B4	Arbour, Zanno & Gates (2016)	Same as Analysis B2, except Paranthodon and Stegosaurus were added as OTUs, and Huayangosaurus removed.	
Analysis B5	Arbour, Zanno & Gates (2016)	Same as Analysis B2, except Paranthodon and Stegosaurus were added as OTUs, in addition to Huayangosaurus.	
Analysis B6	Arbour, Zanno & Gates (2016)	Same as Analysis B5, except Paranthodon was constrained to fall within Ankylosauria.	
Analysis C1	Baron, Norman & Barrett (2017)	Euparkeria used as outgroup. Characters 112, 135, 137, 138, 174 ordered. Anabisetia, Echinodon, Koreanosaurus, Yandosaurus and Yueosaurus removed. ‘New Technology’ search performed with original settings.	
Analysis C2	Baron, Norman & Barrett (2017)	Same as Analysis C1, except Paranthodon was added as an OTU.	
Analysis C3	Baron, Norman & Barrett (2017)	Same as Analysis C2, except Stegosaurus replaced Huayangosaurus as the exemplifier for Stegosauria.	
Analysis C4	Baron, Norman & Barrett (2017)	Same as Analysis C2, except Stegosaurus was added as an OTU, as well as Huayangosaurus.	
Analysis C5	Baron, Norman & Barrett (2017)	Same as Analysis C4, except Isaberrysaura was added as an OTU.	
Analysis C6	Baron, Norman & Barrett (2017)	Same as Analysis C4, except Paranthodon was constrained to fall within Ornithopoda.	
Analysis D1	Boyd (2015)	Marasuchus used as outgroup. All characters unordered. ‘New Technology’ search performed with original settings of Boyd (2015).	
Analysis D2	Boyd (2015)	Same as Analysis D1, except Paranthodon was added as an OTU.	
Analysis D3	Boyd (2015)	Same as Analysis D2, except Huayangosaurus was added as an OTU.	
Analysis D4	Boyd (2015)	Same as Analysis D2, except Stegosaurus was added as an OTU.	
Analysis D5	Boyd (2015)	Same as Analysis D2, except Huayangosaurus and Stegosaurus were added as OTUs.	
Analysis D6	Boyd (2015)	Same as Analysis D5, except Isaberrysaura added as an OTU.	
Analysis D7	Boyd (2015)	Same as Analysis D5, except Paranthodon was constrained to fall within Ornithopoda.	
Analysis D8	Boyd (2015)	Same as Analysis D5, except Paranthodon was constrained to fall within Thyreophora.	
Analysis E1	Raven & Maidment (2017)	Pisanosaurus used as outgroup. The first 24 continuous characters were ordered, as were characters 34, 111 and 112. Discrete characters weighted equally. Character list and character scorings updated from Raven & Maidment (2017).	
Analysis E2	Raven & Maidment (2017)	Same as Analysis E1, except Isaberrysaura added as an OTU	
Analysis E3	Raven & Maidment (2017)	Same as Analysis E1, except Paranthodon was constrained to fall within Ankylosauria.	
Analysis F1	Thompson et al. (2012)	Lesothosaurus used as outgroup. Bissektipelta excluded as an OTU. Characters 25, 27, 32, 133, 159, 167 removed. All remaining characters unordered and equally weighted. ‘Traditional’ search performed with original settings of Thompson et al. (2012).	
Analysis F2	Thompson et al. (2012)	Same as Analysis F1, except that a ‘New Technology’ search was performed and Paranthodon was included as an OTU.	
Analysis F3	Thompson et al. (2012)	Same as Analysis F2, except that Paranthodon was constrained to fall within Stegosauria.	

Baron, Norman & Barrett (2017)

The updated analyses of Baron, Norman & Barrett (2017) were performed with Euparkeria capensis as the outgroup, as in the original analysis. The characters 112, 135, 137, 138 and 174 were ordered and, as in the original analysis, the five unstable taxa Anabisetia saldiviai, Echinodon becklesii, Koreanosaurus boseongensis, Yandosaurus hongheensis and Yueosaurus tiantaiensis were excluded from the analyses. Analysis C1 was produced with the same settings as the original Baron, Norman & Barrett (2017) analysis to make sure the original topology could be replicated. The original analysis used Huayangosaurus as the taxonomic exemplifier for Stegosauria.

Analysis C2 included Paranthodon as an OTU into the original analysis. In Analysis C3, Paranthodon was again included but Stegosaurus replaced Huayangosaurus as the stegosaurian exemplifier. Analysis C4 included Paranthodon, Huayangosaurus and Stegosaurus as OTUs, with the latter two acting as exemplifiers for Stegosauria.

In Analysis C5, the recently described taxon Isaberrysaura (Salgado et al., 2017) was included along with Paranthodon, Huayangosaurus and Stegosaurus. This taxon was included here because although it was recovered as a basal neornithischian by Salgado et al. (2017), it possesses numerous anatomical features normally associated with thyreophorans, and was found to be a stegosaur in Han et al. (2017).

A constraint tree was then written (Analysis C6), using Analysis C4 as a starting point, to test the hypothesis that Paranthodon could be an ornithopod, owing to the similarities of the posterior process of the premaxilla.

Boyd (2015)

Marasuchus lilloensis was used as the outgroup taxon for all analyses of Boyd (2015), and all characters were unordered, as in the original analysis. The original analysis did not include a taxonomic exemplifier for Stegosauria, instead including several basal thyreophorans. Analysis D1 was performed, with no additional taxa included, to make sure the original analysis could be replicated.

In Analysis D2 Paranthodon was added as an OTU to the original analysis. The basal stegosaur Huayangosaurus was then added to the dataset, as well as Paranthodon, so that it included a stegosaurian exemplifier (Analysis D3). Huayangosaurus was then replaced as the exemplifier for Stegosauria by the derived stegosaur Stegosaurus, with Paranthodon also included as an OTU, in Analysis D4.

In Analysis D5, both Huayangosaurus and Stegosaurus were included as exemplifiers for Stegosauria, with Paranthodon also as an OTU.

To again test the systematic positioning of Isaberrysaura, it was added as an OTU to the Boyd (2015) dataset (Analysis D6), along with Paranthodon, Huayangosaurus and Stegosaurus.

Constraint trees were again written to test the lability of Paranthodon, using Analysis D5 as a starting point. Analysis D7 constrained Paranthodon to be within Ornithopoda, and Analysis D8 constrained Paranthodon to be within Thyreophora.

Raven & Maidment (2017)

In Analysis E1, the character list of Raven & Maidment (2017) was updated following a more thorough description of Paranthodon and character scorings were updated to include the dorsal vertebra. Pisanosaurus was used as the outgroup taxon and, as in the original analysis, the 24 continuous characters were ordered, as were the discrete characters 34, 111 and 112. All discrete characters were weighted equally and the continuous characters were automatically rescaled in TNT. In Analysis E2, Isaberrysaura mollensis was also added as an OTU. The full character list and justifications to changes to the original character list can be found in the Supplementary Material.

A constraint tree was then produced with Paranthodon being enforced to fall within Ankylosauria (Analysis E3).

Thompson et al. (2012)

As in the original analysis of Thompson et al. (2012), Lesothosaurus was used as the outgroup, Bissektipelta was excluded as an OTU, the characters 25, 27, 32, 133, 159 and 167 were removed from the analysis and all remaining characters were unordered and equally weighted. Analysis F1 was performed to ensure the original results could be replicated.

Paranthodon was included as an OTU in Analysis F2, with the stegosaurian exemplifiers of Huayangosaurus and Stegosaurus already included in the dataset.

A constraint tree with Paranthodon being enforced into Stegosauria was then produced (Analysis F3).

Results

Arbour & Currie (2016)

The original strict consensus tree of Arbour & Currie (2016; Fig. 11) was replicated in Analysis A1, using the same settings as the original analysis, although this found a tree length of 421 rather than the reported 420; a full list of the results of all analyses can be found in Table 4. Running the analysis of Arbour & Currie (2016) with a ‘New Technology’ search reduced the number of most parsimonious trees (MPTs) from 3,030 in the original analysis to 11 (Analysis A2), with a length of 421. The use of a second, ‘Traditional’, search with TBR branch-swapping on RAM trees was not possible due to computational limits, although this would not change the topology of the strict consensus (Goloboff, Farris & Nixon, 2008). In the strict consensus tree, Nodosauridae had a similar lack of resolution to the original analysis. Gastonia and Ahshislepelta show the same sister taxon relationship basal to Ankylosauridae. Shamosaurinae was found outside of Ankylosaurinae. The rest of Ankylosaurinae had a higher resolution than the strict consensus tree of Arbour & Currie (2016), with Dyoplosaurus found outside of Ankylosaurini. The resolution was as high as that of the 50% majority rule tree of Arbour & Currie (2016).

Table 4 Results of all phylogenetic analyses.

Stegosaurian exemplifier for each analysis is stated, as is the placement of Paranthodon africanus, and any other results of importance.

Analysis	Source of original	Stegosaurian exemplifier	Placement of Paranthodon	Other results	
Analysis A1	Arbour & Currie (2016)	Huayangosaurus	n/a	Same as Arbour & Currie (2016)	
Analysis A2	Arbour & Currie (2016)	Huayangosaurus	n/a	Higher resolution in strict consensus than Arbour & Currie (2016)	
Analysis A3	Arbour & Currie (2016)	Huayangosaurus	Ankylosaur	9 MPTs	
Analysis A4	Arbour & Currie (2016)	Stegosaurus	Base of Thyreophora	8 MPTs and increased resolution	
Analysis A5	Arbour & Currie (2016)	Huayangosaurus and Stegosaurus	Stegosaur	9 MPTs and increased resolution	
Analysis A6	Arbour & Currie (2016)	Huayangosaurus and Stegosaurus	Ankylosaur (constrained)	9 MPTs and reduced resolution.	
Analysis B1	Arbour, Zanno & Gates (2016)	Huayangosaurus	n/a	Same as Arbour, Zanno & Gates (2016)	
Analysis B2	Arbour, Zanno & Gates (2016)	Huayangosaurus	n/a	Higher resolution in strict consensus than Arbour, Zanno & Gates (2016)	
Analysis B3	Arbour, Zanno & Gates (2016)	Huayangosaurus	Nodosaur	3 MPTs and increased resolution in Nodosauridae	
Analysis B4	Arbour, Zanno & Gates (2016)	Stegosaurus	Base of Thyreophora	5 MPTs and increased resolution in Ankylosauridae	
Analysis B5	Arbour, Zanno & Gates (2016)	Huayangosaurus and Stegosaurus	Stegosaur	2 MPTs and similar resolution	
Analysis B6	Arbour, Zanno & Gates (2016)	Huayangosaurus and Stegosaurus	Ankylosaur (constrained)	3 MPTs and similar resolution	
Analysis C1	Baron, Norman & Barrett (2017)	Huayangosaurus	n/a	Same as Baron, Norman & Barrett (2017)	
Analysis C2	Baron, Norman & Barrett (2017)	Huayangosaurus	Ankylosaur	Little resolution	
Analysis C3	Baron, Norman & Barrett (2017)	Stegosaurus	Stegosaur	Higher resolution	
Analysis C4	Baron, Norman & Barrett (2017)	Huayangosaurus and Stegosaurus	Stegosaur	Very high resolution	
Analysis C5	Baron, Norman & Barrett (2017)	Huayangosaurus and Stegosaurus	Stegosaur	Little resolution and Isaberrysaura= ornithopod	
Analysis C6	Baron, Norman & Barrett (2017)	Huayangosaurus and Stegosaurus	Ornithopod (constrained)	Severely reduced resolution in Ornithopoda	
Analysis D1	Boyd (2015)	n/a—Scelidosaurus most derived thyreophoran	n/a	Same as Boyd (2015)	
Analysis D2	Boyd (2015)	n/a—Scelidosaurus most derived thyreophoran	Base of Ornithischia	Thyreophora basal to Heterodontosauridae, Marginocephalia basal to Ornithopoda	
Analysis D3	Boyd (2015)	Huayangosaurus	Ornithopod, sister-taxon to Huayangosaurus	Huayangosaurus= ornithopod and reduced resolution in Ornithopoda	
Analysis D4	Boyd (2015)	Stegosaurus	Ornithopod, sister- taxon to Stegosaurus	Stegosaurus= ornithopod and increased resolution	
Analysis D5	Boyd (2015)	Huayangosaurus and Stegosaurus	Ornithopod, sister- taxon to Huayangosaurus and Stegosaurus	Huayangosaurus and Stegosaurus= ornithopod and little resolution	
Analysis D6	Boyd (2015)	Huayangosaurus and Stegosaurus	Ornithopod, sister- taxon to Huayangosaurus and Stegosaurus	Huayangosaurus and Stegosaurus= ornithopod and little resolution. Isaberrysaura= ornithopod	
Analysis D7	Boyd (2015)	Huayangosaurus and Stegosaurus	Ornithopod (constrained)	Huayangosaurus and Stegosaurus outside of Ornithischia and increased resolution in Ornithopoda.	
Analysis D8	Boyd (2015)	Huayangosaurus and Stegosaurus	Thyreophoran	Ornithopoda resolution increased, Thyreophora resolution decrease	
Analysis E1	Raven & Maidment (2017)	n/a	Stegosaur	Similar to Raven & Maidment (2017)	
Analysis E2	Raven & Maidment (2017)	n/a	Eurypodan	Isaberrysaura= basal stegosaur. Reduced resolution in Eurypoda	
Analysis E3	Raven & Maidment (2017)	n/a	Ankylosaur (constrained)	Reduced resolution in Ankylosauria	
Analysis F1	Thompson et al. (2012)	Huayangosaurus and Stegosaurus	n/a	Same as Thompson et al. (2012)	
Analysis F2	Thompson et al. (2012)	Huayangosaurus and Stegosaurus	Ankylosaur	Higher resolution in strict consensus than Thompson et al. (2012)	
Analysis F3	Thompson et al. (2012)	Huayangosaurus and Stegosaurus	Stegosaur (constrained)	Resolution of Nodosauridae increased	

When Paranthodon was added as an OTU and Huayangosaurus was used as the only stegosaurian exemplifier, as in the original analysis, (Analysis A3), eight MPTs were recovered with a length of 424. Paranthodon was recovered as an ankylosaur, in a polytomy basal to Ankylosaurinae with Gobisaurus and Shamosaurus.

When the more derived stegosaur Stegosaurus was used as the stegosaurian exemplifier, and Huayangosaurus excluded as an OTU (Analysis A4), eight MPTs were recovered with a length of 425. The strict consensus tree had a similar topology to Analysis A2, however Paranthodon was found in a polytomy with Stegosaurus and Kunbarrasaurus near the base of Thyreophora.

In Analysis A5, both Huayangosaurus and Stegosaurus were used as exemplifiers for Stegosauria, and Paranthodon was included as an OTU. This produced nine most parsimonious trees of length 427 and again had high resolution throughout the strict consensus tree. Stegosauria formed a monophyletic group, with Huayangosaurus basal to a sister-taxon relationship between Parathodon and Stegosaurus. Kunbarrasaurus was found at the base of Ankylosauria again.

Analysis A6 constrained Paranthodon to be an ankylosaur. This produced nine most parsimonious trees, of length 428, with slightly reduced resolution in Ankylosauridae, in comparison to the unconstrained tree of Analysis A5. Paranthodon was found at the base of Ankylosauridae in a polytomy with Shamosaurus scutatus and Gobisaurus domoculus. The constraint tree was analysed using the Templeton Test, which indicated the length differences between the unconstrained tree and the constrained tree were non-significant.

Arbour, Zanno & Gates (2016)

The original settings of Arbour, Zanno & Gates (2016) were replicated in Analysis B1 and the same results were found. Running the analysis with a ‘New Technology’ search (Analysis B2) produced three MPTs of length 551. The use of a second, ‘Traditional’, search with TBR branch-swapping on RAM trees was not possible due to computational limits, although this would not change the topology of the strict consensus (Goloboff, Farris & Nixon, 2008). The strict consensus had higher resolution than that of the original analysis, approaching that of the 50% majority rule tree, particularly within Ankylosauridae.

When Paranthodon was added as an OTU and Huayangosaurus was used as the only stegosaurian exemplifier, as in the original analysis (Analysis B3), three MPTs were found, of length 555. Paranthodon was recovered as a basal nodosaur and there was reduced resolution in Ankylosauridae relative to Analysis B2, but increased resolution within Nodosauridae, including a monophyletic Struthiosaurus.

In Analysis B4, the more derived stegosaur Stegosaurus was used as the stegosaurian exemplifier and Huayangosaurus was excluded as an OTU. This resulted in five MPTs of length 554. The strict consensus had a similar resolution within Nodosauridae to Analysis B3 but there was increased resolution in Ankylosauridae. Paranthodon was found as a sister-taxon to Stegosaurus as the base of Thyreophora.

When Paranthodon was added as an OTU and both Huayangosaurus and Stegosaurus were used as the stegosaurian exemplifiers (Analysis B5), two MPTs of length 557 were found. Stegosauria was monophyletic, with Huayangosaurus basal to a sister-taxon relationship between Paranthodon and Stegosaurus. There was similar high resolution in Ankylosauridae relative to Analysis B4 but there was reduced resolution within Nodosauridae.

Analysis B6 constrained Paranthodon to be an ankylosaur. This produced three MPTs, of length 558, with similar resolution in both Ankylosauridae and Nodosauridae relative to Analysis B5. Paranthodon was found as a sister-taxon to Shamosaurus and Gobisaurus within Ankylosauridae. The constraint tree was analysed using the Templeton Test, which indicated the length differences between the unconstrained tree and the constrained tree were non-significant.

Baron, Norman & Barrett (2017)

The original settings of the basal ornithischian analysis of Baron, Norman & Barrett (2017) were replicated and the same topology was found (Analysis C1).

The dataset was then updated to include Paranthodon as an OTU, and Huayangosaurus was used as the exemplifier for Stegosauria, as in the original analysis (Analysis C2). The ‘New Technology’ search followed by TBR branch-swapping resulted in 144 most parsimonious trees of length 583; however, the strict consensus tree provided little resolution. A 50% majority rule tree suggested Paranthodon might be closer related to Ankylosauria than to Huayangosaurus.

The original exemplifier for Stegosauria, Huayangosaurus, was then replaced by Stegosaurus, and Paranthodon was included as an OTU (Analysis C3). This produced 96 most parsimonious trees of length 583 and the strict consensus provided much higher resolution throughout the tree than in Analysis C2. Paranthodon was found as sister-taxon to Stegosaurus, with Ankylosauria a separate lineage within Thyreophora.

In Analysis C4, both Huayangosaurus and Stegosaurus were included as exemplifiers for Stegosauria, and Paranthodon was included as an OTU. This produced 84 most parsimonious trees of length 587 and very high resolution in the strict consensus. Stegosauria was found to be monophyletic, with Paranthodon more closely related to Stegosaurus than to Huayangosaurus.

Analysis C5 included the newly described Isaberrysaura as an OTU, in addition to Paranthodon, Huayangosaurus and Stegosaurus. This produced 340 most parsimonious trees of length 605, and little resolution in the strict consensus tree in Ornithopoda, but Thyreophora had the same topology as Analysis C4. Isaberrysaura was found in a large polytomy within Ornithopoda.

Analysis C6 constrained Paranthodon to Ornithopoda. This resulted in 10 most parsimonious trees of length 595. Relative to the unconstrained Analysis C4, this increased the resolution in Heterodontosauridae slightly but caused a severe reduction in resolution in Ornithopoda; Paranthodon was found in a polytomy at the base of the group with 11 other taxa. Again, the use of the Templeton Test showed that the differences between the unconstrained tree and the constrained tree were non-significant.

Boyd (2015)

The original results of the basal ornithischian phylogeny of Boyd (2015) were replicated here, using the same search settings (Analysis D1).

The dataset was then updated to include Paranthodon as an OTU (Analysis D2), with Scelidosaurus the most derived thyreophoran included from the original dataset. The use of a second, ‘Traditional’, search with TBR branch-swapping on RAM trees was not possible due to computational limits, although this would not change the topology of the strict consensus (Goloboff, Farris & Nixon, 2008). The ‘New Technology’ search produced two most parsimonious trees of length 884. In the strict consensus tree, Paranthodon was found to be in a sister-taxon relationship with Pisanosaurus. Interestingly, Thyreophora was basal to Heterodontosauridae, and Marginocephalia was basal to Ornithopoda.

In Analysis D3, Huayangosaurus was included to act as a stegosaur exemplifier, and Paranthodon was also added as an OTU. This produced five most parsimonious trees, of length 921, and there was reduced resolution in the strict consensus. Paranthodon and Huayangosaurus were found as sister-taxa at the base of Iguanodontia, distant from the other taxa that traditionally comprise Thyreophora.

Huayangosaurus was then replaced as the stegosaurian exemplifier by Stegosaurus, with Paranthodon again included as an OTU (Analysis D4). This produced three most parsimonious trees, of length 928. The strict consensus tree had increased resolution relative to Analysis D3, and Paranthodon and Stegosaurus were found as sister-taxa within Ornithopoda, again distant from Thyreophora.

In Analysis D5, both Huayangosaurus and Stegosaurus were used as the exemplifiers for Stegosauria, and Paranthodon was included as an OTU. This produced seven most parsimonious trees of length 955, but with a reduced resolution in most of the tree. Paranthodon, Huayangosaurus and Stegosaurus were found as sister-taxa, again separate from Thyreophora.

Isaberrysaura was then included, as well as Huayangosaurus, Stegosaurus and Paranthodon, into Analysis D6. Five most parsimonious trees, of length 968, were produced. There was again little resolution in the strict consensus, particularly in Neornithischia, with Isaberrysaura, Huayangosaurus, Stegosaurus and Paranthodon forming part of a large polytomy at the base.

Analysis D7 constrained Paranthodon within Ornithopoda. This produced six most parsimonious trees of length 964, and increased resolution in Ornithopoda relative to the unconstrained Analysis D5. However, Stegosaurus and Huayangosaurus moved out of Ornithischia, as they were not constrained to be within Ornithopoda. Paranthodon was found in a large polytomy at the base of Ornithopoda with nine other taxa.

Analysis D8 constrained Paranthodon, Huayangosaurus and Stegosaurus to Thyreophora. This produced four most parsimonious trees of length 965. The strict consensus had higher resolution in Ornithopoda, but the resolution in Thyreophora was reduced. Paranthodon, Huayangosaurus and Stegosaurus formed a polytomy within Thyreophora. Stormbergia dangershoeki, a taxon that Baron, Norman & Barrett (2017) have recently synonymised with Lesothosaurus, moved to within Thyreophora in this analysis. The Templeton Test again showed that the differences between the unconstrained trees and the constrained trees were all non-significant.

Raven & Maidment (2017)

The most recent phylogeny of Stegosauria by Raven & Maidment (2017) showed Paranthodon and Tuojiangosaurus to clade together, a result that was found again here in the one most parsimonious tree of length 279.65 (Analysis E1). Isaberrysaura, the Argentinian dinosaur found as a neornithischian by Salgado et al. (2017), was then found in a sister-taxon relationship with Gigantspinosaurus (Analysis E2). However, the strict consensus of the four most parsimonious trees of length 285.38 had a lack of resolution at the base of Eurypoda. Analysis E3 was produced to constrain Paranthodon to within Ankylosauria, using Analysis E1 as a starting point. This produced one most parsimonious tree of length 280.43, 0.78 steps longer than Analysis E1. The Templeton Test showed that there were no significant difference between the constrained and the unconstrained trees in all analyses.

Thompson et al. (2012)

Using the original settings of Thompson et al. (2012), the original results were replicated (Analysis F1).

The dataset was then updated to include Paranthodon as an OTU (Analysis F2), using both Huayangosaurus and Stegosaurus as the exemplifiers for Stegosauria, as in the original analysis. This analysis, using a ‘New Technology’ search, produced five MPTs with a length of 529, although the use of a second, ‘Traditional’, search with TBR branch-swapping on RAM trees was not possible due to computational limits, although this would not change the topology of the strict consensus (Goloboff, Farris & Nixon, 2008). The results vastly improved on the 4,248 MPTs with a length of 527 produced in the ‘Traditional’ searches of the original analysis, and there was an improvement in the resolution of the strict consensus tree, especially within Ankylosauridae, where it approaches the resolution of the 50% majority rule tree of Thompson et al. (2012). Pinacosaurus was found to be paraphyletic; Pinacosaurus mephistocephalus and Dyopolosaurus acutosquameus are sister-taxa, as are Pinacosaurus grangeri and Minotaurasaurus ramachandrani. Ankylosaurus magniventris and Euoplocephalus tutus are also found as sister-taxa. Stegosaurus and Huayangosaurus clade together to form Stegosauria, which was sister taxon to Ankylosauria. Paranthodon was found in a large polytomy at the base of Ankylosauria.

Analysis F3 constrained Paranthodon to Stegosauria. This produced three most parsimonious trees of length 531, two steps longer than the unconstrained Analysis F1. The resolution of Ankylosauridae did not change but the resolution of Nodosauridae increased. Paranthodon had a closer relationship to Stegosaurus than to Huayangosaurus. Again, there were no significant differences between the constrained and the unconstrained trees according to the Templeton Test.

Discussion

The use of basal exemplifiers in cladistic analysis

When Paranthodon was added as an OTU to the dataset of Arbour & Currie (2016) and Huayangosaurus used as the stegosaurian exemplifier (Analysis A3), Paranthodon was found as an ankylosaur. However, when the exemplifier was changed to Stegosaurus (Analysis A4), Paranthodon was found at the base of Thyreophora. When both Huayangosaurus and Stegosaurus were included in the analysis, Stegosauria became monophyletic with Huayangosaurus basal to Paranthodon + Stegosaurus (Analysis A5).

Similarly, when Huayangosaurus was used as the stegosaurian exemplifier and Paranthodon was added as an OTU into the dataset of Arbour, Zanno & Gates (2016), Paranthodon was found as a basal nodosaur (Analysis B3). However, Paranthodon was found at the base of Thyreophora when the stegosaurian exemplifier was changed to Stegosaurus (Analysis B4). Paranthodon was then found in a monophyletic Stegosauria when both Huayangosaurus and Stegosaurus were included in the analysis (Analysis B5). The inclusion of Paranthodon into the Baron, Norman & Barrett (2017) dataset reduced the resolution of the tree, but a 50% majority rule tree found Paranthodon as an ankylosaur (Analysis C2). When Stegosaurus replaced Huayangosaurus as the stegosaurian exemplifier (Analysis C3), the resolution in the tree increased and Paranthodon was sister-taxon to Stegosaurus. When both Huayangosaurus and Stegosaurus were included in the analysis (Analysis C4), there was again increased resolution and a monophyletic Stegosauria, including Paranthodon.

The inclusion of Paranthodon to the Boyd (2015) dataset (Analysis D2) found Paranthodon as a basal ornithischian, sister-taxon to Pisanosaurus, with large topological changes in the rest of the tree. When Huayangosaurus was included as an OTU (Analysis D3), Paranthodon and Huayangosaurus were sister-taxa within Ornithopoda. Replacing Huayangosaurus as the stegosaurian exemplifier with Stegosaurus (Analysis D4) improved the resolution of the tree but again both Stegosaurus and Paranthodon were found within Ornithopoda.

These results demonstrate that the systematic position of Paranthodon is highly dependent on the clade exemplifier used. When a basal exemplifier is used, Paranthodon is generally found to be an ankylosaur, but resolution is lost. When a more derived exemplifier (Stegosaurus) is used, Paranthodon is found as a stegosaur. When both a basal and a derived exemplifier is used, Paranthodon is found as a stegosaur, Stegosauria is found to be monophyletic, and resolution of the entire tree is generally increased (Fig. 7). This indicates that the choice of exemplifier as a basal taxon within a clade may be inappropriate if the aim of the analysis is to test the phylogenetic position of a taxon that potentially shows more derived characteristics of a clade. This contrasts with most literature on the subject (e.g., Yeates, 1995; Griswold et al., 1998; Prendini, 2001; Brusatte, 2010), which argues that an exemplifier species should be a basal taxon within its respective clade.

Figure 7 Analyses of Arbour, Zanno & Gates (2016) (A, B) and Baron, Norman & Barrett (2017) (C, D) showing labile positioning of Paranthodon depending on stegosaurian exemplifier used.

Analysis B3 and C2 use Huayangosaurus as stegosaurian exemplifier for analyses of Arbour, Zanno & Gates (2016) and Baron, Norman & Barrett (2017), respectively. Analysis B4 of Arbour, Zanno & Gates (2016) uses Stegosaurus as stegosaurian exemplifier, and Analysis C3 of Baron, Norman & Barrett (2017) uses both Huayangosaurus and Stegosaurus. Paranthodon is found as a basal nodosaurid in B3, in a large polytomy in C2, as a basal thyreophoran in B4 and in a monophyletic Stegosauria in C3. Resolution of analyses increases when derived taxonomic exemplifiers are used.

A more robust approach would be to use multiple exemplifiers, and this method has been argued previously (Prendini, 2001; Brusatte, 2010), but is not common practice. The use of supraspecific taxa to represent groups of species, in any method, can result in changes to topology of a phylogeny when compared to a complete species level analysis (Bininda-Emonds, Bryant & Russell, 1998), even the use of multiple exemplifiers. While the use of exemplifiers can produce accurate tree topologies that are subsequently and independently found in later analyses (for example, Butler, Upchurch & Norman, 2008), caution should be applied when interpreting the phylogenies (Spinks et al., 2013), especially when including the use of fragmentary material. The ability of ‘New Technology’ searches in TNT to analyse large datasets in less time than ‘Traditional’ searches (Goloboff, Farris & Nixon, 2008) means more taxa can be included in the analysis, which would increase the accuracy dramatically (Prendini, 2001). This means it is not always impractical to include each species as a separate terminal. Phylogenetic super-matrices (Gatesy et al., 2002) therefore could and should be implemented to analyse evolutionary relationships, meaning the use of exemplifiers would be redundant.

Figure 8 Strict consensus tree from Analysis D6; inclusion of Paranthodon, Huayangosaurus, Stegosaurus and Isaberrysaura as OTUs into the Boyd (2015) dataset.

Only two synapomorphies characterise the group of basal thyreophorans; a ridge on the lateral surface of surangular, which is not present in stegosaurs, and a concave lingual surface of maxillary teeth, which is not a eurypodan character. This demonstrates that the Boyd (2015) dataset is inadequate for accurately testing the position of eurypodans, possibly explaining the positioning of Isaberrysaura as an ornithopod in Salgado et al. (2017).

That basal exemplifiers may be inappropriate is further supported by our analyses of the Boyd (2015) dataset. The recently described taxon Isaberrysaura (Salgado et al., 2017) was included as an OTU in Analysis D6, as well as Huayangosaurus, Stegosaurus and Paranthodon (Fig. 8). This taxon was included here because although it was recovered as a basal neornithischian by Salgado et al. (2017), it possesses numerous anatomical features normally associated with thyreophorans, and was found to be a stegosaur in Han et al. (2017). Analysis D6 resulted in Isaberrysaura being found as a basal neornithischian, along with Paranthodon and the unambiguous stegosaurs Huayangosaurus and Stegosaurus. This surprising result is an artefact of the character distribution of the Boyd (2015) dataset; there are only seven characters that unite either Eurypoda, Eurypoda + Alcovasaurus, or Stegosauria in the Raven & Maidment (2017) dataset that are found in the Boyd (2015) dataset, equating to 2.7% of the total number of characters. Additionally, there are only two synapomorphies that unite the taxa used to represent Thyreophora (i.e., Lesothosaurus, Scutellosaurus, Emausaurus and Scelidosaurus) in the Boyd (2015) dataset; character 86: a strong, anteroposteriorly extending ridge present on the lateral surface of the surangular, and character 122: a concave lingual surface of maxillary teeth. These features, although synapomorphies for basal thyreophorans, are lost in stegosaurs and ankylosaurs, and this suggests the Boyd (2015) dataset cannot adequately test the relationships of eurypodans. The placement of Isaberrysaura as a basal neornithischian in Salgado et al. (2017) is almost certainly due to the fact that the dataset of Boyd (2015) does not contain the character data required to rigorously test the phylogenetic position of taxa which may be derived members of clades. It is therefore likely that, as found by Han et al. (2017), Isaberrysaura is a member of the Thyreophora.

The anatomy of Paranthodon is enigmatic, with features similar to many other members of Ornithischia. The tooth morphology and the presence of a secondary maxillary palate is reminiscent of ankylosaurs, and the cingulum is widely distributed among ornithischians, as is the sinuous curve of the anterior process of the premaxilla (Butler, Upchurch & Norman, 2008). The robust posterior process of the premaxilla is similar to that of ornithopods. The triangular maxilla in lateral view is a feature seen widely across Thyreophora, and an edentulous premaxilla is common to most stegosaurs but also many other derived ornithischians. There are no features of the skull that unite Paranthodon firmly within Stegosauria and Paranthodon contains no synapomorphies that place it unequivocally within Stegosauria. However, the orientation of the transverse processes of the mid-dorsal vertebra at higher than 50 degrees to the horizontal was considered a synapomorphy of the clade by Galton & Upchurch (2004), and this condition is present in Paranthodon. The discovery of a well-preserved specimen of Stegosaurus (Maidment, Brassey & Barrett, 2015) showed the transverse processes of the dorsal vertebrae vary in projection angle down the vertebral column. This character statement cannot, therefore, be used as a synapomorphy of the group; however, the condition is present in all stegosaurs with dorsal vertebrae known, other than Gigantspinosaurus.

On the available evidence, both anatomical and phylogenetic, it appears the most parsimonious solution is to refer Paranthodon to Stegosauria. The general anatomy appears most similar to the stegosaurs Tuojiangosaurus and Stegosaurus, and numerous phylogenetic analyses indicate, when both basal and derived exemplifiers are used, that there is a close relationship between Paranthodon and Stegosaurus. The increased resolution afforded by the use of Stegosaurus suggests some character conflict is being resolved, and the relative instability when Huayangosaurus is used could be because of symplesiomorphies between basal ankylosaurs and basal stegosaurs preventing a more derived taxon from ‘finding a place’ in the tree.

The use of constraint trees also provides evidence for Paranthodon as a stegosaur, although the use of the Templeton Test shows alternative hypotheses cannot be ruled out. Constraining Paranthodon to within Ankylosauria in Analysis A6 of Arbour & Currie (2016) reduced the resolution in Ankylosauridae and increased the number of steps in the tree. Similarly, constraining Paranthodon to within Ankylosauria in Analysis B6 of Arbour, Zanno & Gates (2016) increased the number of steps in the tree and the number of most parsimonious trees found. In Analysis C6, where Paranthodon was constrained to within Ornithopoda, there was a reduced resolution within Ornithopoda and an increased number of steps in the tree. In Analysis D7 of the Boyd (2015) dataset, where Paranthodon was constrained within Ornithopoda, Stegosauria moved outside of Ornithischia and the number of steps in the tree increased, although there was increased resolution in Ornithopoda (as Stegosaurus and Huayangosaurus had moved out of the group). Constraining Paranthodon within Thyreophora using the Boyd (2015) dataset (Analysis D8) increased the resolution in Ornithopoda, but reduced it in Thyreophora, and there were more steps in the tree. However, Stormbergia dangershoeki, a taxon that was synonymised with Lesothosaurus diagnosticus by Baron, Norman & Barrett (2017), moved into Thyreophora. Constraining Paranthodon to be an ankylosaur in the updated dataset of Raven & Maidment (2017) (Analysis E3) increased the tree length of the one most parsimonious tree. In Analysis F3, where Paranthodon was constrained within Stegosauria using the Thompson et al. (2012) dataset, the resolution of Nodosauridae increased, although the tree length also increased. Although there is a lot of evidence from constraint trees for the positioning of Paranthodon within Stegosauria, it is also shown to be labile within Thyreophora. This labile positioning is likely to be due to both deep-rooted homology between Stegosauria and Ankylosauria, given the close evolutionary relationships of the two lineages of Thyreophora, as well as convergent evolution, given the similar ecology of the two groups of animals.

The placing of Paranthodon within Stegosauria means that the presence of the medial maxillary process is autapomorphic and evolved independently in stegosaurs and ankylosaurs. Paranthodon is thus a valid genus. However, the systematic positioning of Paranthodon is likely to stay labile unless more material is found, and until a thyreophoran or ornithischian super-matrix can be utilised for phylogenetic analyses.

Importance of Paranthodon

The results presented here suggest that Paranthodon is most robustly recovered as a stegosaur and this has important implications for this iconic yet surprisingly poorly understood group of dinosaurs. Paranthodon is one of the youngest stegosaurs and stratigraphically close to the assumed extinction event of the group (Pereda Suberbiola et al., 2003). There are few other pieces of evidence for Cretaceous stegosaurs; Stegosaurus homheni was found in the Lower Cretaceous of Inner Mongolia (Maidment et al., 2008) and the Burgos specimen of Dacentrurus armatus was found in the Lower Cretaceous of Spain (Pereda Suberbiola et al., 2003; Maidment et al., 2008). Additionally, indeterminate stegosaurians have been identified in the Lower Cretaceous of Inner Mongolia (previously known as Wuerhosaurus ordosensis; Maidment et al., 2008) and the Early Cretaceous of Portugal (Pereda Suberbiola et al., 2005). Stegosaurian ichnofacies have also reportedly been identified in the Early Cretaceous of China (Xing et al., 2013) (although these appear similar to sauropod footprints according to Salisbury et al. (2016)) and in the Lower Cretaceous Broome Sandstone of Western Australia (Salisbury et al., 2016), as well as in the Upper Cretaceous of Southern India (Galton & Ayyasami, 2017).

The biogeographical distribution of stegosaurs is also quite limited; other than Paranthodon, Kentrosaurus from Tanzania is the only other confirmed occurrence of Stegosauria in Gondwana. The aforementioned Isaberrysaura from Patagonia has characteristics of both basal thyreophorans and basal stegosaurs; however, further study and a postcranial description of the skeleton, are needed to elucidate the taxonomic status of the specimen. Stegosaurian ichnofacies are also reported throughout Gondwana, in Western Australia (Salisbury et al., 2016), Southern India (Galton & Ayyasami, 2017), and Bolivia ( Apesteguia & Gallina, 2011). Additionally, an indeterminate stegosaurian specimen was reported by Haddoumi et al. (2016) in Morocco, and there have been repeated reports to a taxon previously referred to as Dravidosaurus in Southern India (Galton & Ayyasami, 2017).

Paranthodon is therefore an important data point for future evaluations of both the stratigraphic and biogeographic evolution of the clade Stegosauria, as well as for total-group evaluations of Thyreophora.

Phylogeny of Ankylosauria

The recent phylogeny of the ankylosaurian dinosaurs by Arbour & Currie (2016) was re-analysed herein with a ‘New Technology’ search in TNT (Analysis A2). This has improved the resolution of the analysis, especially the relationships of derived ankylosaurids, and reduced the number of MPTs from 3,030 to 11, relative to the original analysis by Arbour & Currie (2016). The resolution of the strict consensus tree in this study is similar to that of the 50% majority rule tree in Arbour & Currie (2016), but Crichtonpelta has moved outside of Ankylosaurinae, meaning it is not the oldest known ankylosaurine. A similar result occurred when running the dataset of Arbour, Zanno & Gates (2016) with a ‘New Technology’ search (Analysis B2); the resolution of Ankylosauridae in the strict consensus improved such that it approached that of the 50% majority rule tree in the original analysis. Additionally, running the ankylosaurian dataset of Thompson et al. (2012) with a ‘New Technology’ search (Analysis F2) improved the resolution of Ankylosauridae in the strict consensus so that it was approaching the resolution of the 50% majority rule tree in the original analysis, which was performed with a ‘Traditional’ search.

The results of these analyses are, therefore, more robust, as the use of strict consensus trees is a more rigorous method than majority rule trees for summarising the information found within the MPTs (Bryant, 2003). This improved resolution is due to the use of ‘New Technology’ searches, rather than the ‘Traditional’ search option used in the original analysis. ‘Traditional’ searches are heuristic, and can get stuck on local parsimony optimums within treespace, whereas ‘New Technology’ searches employ algorithms (Ratchet, Sectorial, Drift and Tree Fusing) that allow more rigorous searches for improved tree scores and a reduced number of optimal trees, within minimal time (Goloboff, Farris & Nixon, 2008). These are much more effective than branch-swapping methods, especially for datasets with hundreds of characters and a large number of taxa.

Conclusions

Our results demonstrate that the use of basal exemplifiers in cladistic analysis may prevent the correct phylogenetic position of derived taxa from being established. Instead, we recommend the use, minimally, of a basal and derived exemplifier for each clade. The phylogenetic position of Paranthodon is highly labile and is dramatically affected by the choice of taxonomic exemplifier, and further material of this enigmatic taxon is required to fully assess its affinities. However, based on the currently available data, it seems most likely that the taxon is a stegosaur.

Supplemental Information

Supplemental Information 1 Online Supplementary Material

Includes results of all analyses, the character list for the updated Raven & Maidment (2017) analyses and the characters from Boyd (2015) that unite either Eurypoda, Eurypoda + Alcovasaurus or Stegosauria in Raven & Maidment (2017).

Click here for additional data file.

Supplemental Information 2 Arbour & Currie (2016) character taxon matrix

Click here for additional data file.

Supplemental Information 3 Arbour, Zanno & Gates (2016) character taxon matrix

Click here for additional data file.

Supplemental Information 4 Boyd (2015) character taxon matrix

Click here for additional data file.

Supplemental Information 5 Baron, Norman & Barrett (2017) character taxon matrix

Click here for additional data file.

Supplemental Information 6 Raven & Maidment (2017) character taxon matrix

Click here for additional data file.

Supplemental Information 7 Thompson et al. (2012) character taxon matrix

Click here for additional data file.

Sandra Chapman and Prof. Paul Barrett (Natural History Museum) provided access to specimens in their care. Harry Taylor (Natural History Museum Photographic Unit) provided photographs of specimens. This work benefitted from discussion with members of the Imperial College Palaeobiology Research Group. Thanks to Kristina Kareh for help with CT-scanning at Imperial College. The Willi Hennig Society sponsored the development and free distribution of TNT. Alexander Schmidt-Lebuhn (Centre for Australian National Biodiversity Research) provided the script for running the Templeton Test in TNT. Comments from Andrew Farke (editor), Jim Kirkland, Victoria Arbour and one anonymous reviewer improved this manuscript and are gratefully acknowledged.

Additional Information and Declarations

Competing Interests

Author Contributions

The authors declare there are no competing interests.

Thomas J. Raven conceived and designed the experiments, performed the experiments, analyzed the data, contributed reagents/materials/analysis tools, prepared figures and/or tables, authored or reviewed drafts of the paper, approved the final draft.

Susannah C.R. Maidment conceived and designed the experiments, analyzed the data, contributed reagents/materials/analysis tools, authored or reviewed drafts of the paper, approved the final draft.

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
