# Peer review of "The systematic position of the enigmatic thyreophoran dinosaur Paranthodon africanus, and the use of basal exemplifiers in phylogenetic analysis"

_PeerJ, doi:10.7717/peerj.4529_

## Round 0.1 · original submission · Minor Revisions

This paper is a well constructed and highly timely reassessment of the important Gondwanan taxon Paranthodon africanus. The reviewers and I are in agreement that only minor changes are required prior to acceptance. These are summarized below--please see the section for each individual reviewer for detailed remarks.

SUMMARY

- As suggested by one reviewer, please provide a bit more information on the association of the vertebral material with the skull. Is the preservation consistent in a way that allows confident association? Are there field notes to support this?
- Also as noted, please ensure that you are fully exploring tree space within your analyses in TNT.
- Other minor comments from the reviewers should be addressed as appropriate.

COMMENTS FROM EDITOR

- This is a bit of a pedantic point, but the introduction states that Galton and Upchurch produced the first cladistic analysis of Stegosauria. As summarized in Maidment et al. 2008, there were several prior analyses (e.g., Carpenter et al., 2001) that also addressed stegosaur phylogeny.
- Some basic measurements should be provided, either as a table or in the text -- e.g., tooth dimensions, tooth row length as preserved, maxilla height, denticle density/mm on teeth, etc.

Reviewer 1 ·

Basic reporting

The manuscript is generally clear and well written, appropriately structured, with sufficient background information and figures. Raw data has been uploaded.

Experimental design

The article represents a piece of original primary research, and addresses a well-defined question (the phylogenetic position of the dinosaur taxon Paranthodon). Methods are generally appropriate and well-conducted, with some minor concerns (see comments for the author, below).

Validity of the findings

Results generally appear well supported and are adequately and carefully discussed, with some minor reservations (see general comments to the author, below).

Additional comments

This paper aims to reassess the phylogenetic position of Paranthodon, a putative stegosaur dinosaur from the Cretaceous of South Africa. It provides a redescription of the type material, and tests the phylogenetic position is a large number of different datasets. It concludes that a stegosaur identity is most likely, but also provides an excellent case study of the importance of including multiple species-level exemplars to represent major clades in phylogenetic analyses, rather than just a single basal taxon. I recommend publication with some minor revisions, and have only a couple of substantial concerns, around the association of the vertebra with the rest of the type material, and the conclusions drawn about ankylosaur phylogeny. These and additional points are explored in more detail below:

Line 32: ‘parasagittal’ is misspelled

Lines 46–48: You state that ‘The lithologic description of the upper unit by McPhee et al. (2016) matches the matrix of NHMUK R47338, and thus it is likely that Paranthodon is derived from this unit.’ I think this statement should be unpacked in a bit more detail. What is the matrix present with NHMUK R47338, and how exactly does it compare with that of the upper member of the Kirkwood?

Line 88: You state that the holotype includes a dorsal vertebra, and elsewhere say that this was mentioned by Galton & Coombs. I have checked Galton & Coombs, and they do not mention any vertebra as part of the holotype material of Paranthodon africanus, but they do mention ‘three vertebrae in matrix’ with the specimen number NHMUK [BMNH] 47337a, which is similar to the specimen number for the holotype of the pareiasaur Anthodon serrianus (holotype NHMUK 47337). Owen also does not mention vertebral material as part of the same specimen as the type of Paranthodon africanus. I think you need to provide more details on this vertebral fragment: has it been mentioned in the literature previously, is it really from the same locality as the holotype etc.? I am slightly concerned about the association, given that Owen mixed together Permian and Cretaceous material when describing Anthodon serrianus, and given that this vertebra appears crucial to determining the phylogenetic position. I don't know enough about pareiasaur vertebrae to draw any comparisons with them.

Lines 90–94: You list the teeth as referred to Paranthodon, but at the same time say that they are indeterminate thyreophoran remains. These two statements are incompatible. Do you mean to say that the teeth are “previously referred specimens”?

Lines 114–115: Galton & Coombs (1981: fig. 1A) show a large posterior process of the premaxilla, and also describe it on page 301. Are you to correct to say that they misidentified this process as the nasal?

Line 154: is Butler, Porro and Norman the correct citation for an observation of Hypsilophodon?

Line 156: you need a full stop or semi-colon after “Leahey et al., 2015)”

Line 312: Same comment as above – call them “previously referred specimen”

Line 317: Details of the CT scanning should be provided in a methods section. What was scanned, where was it scanned, and what settings were used? What software was used to analyse the data, and where is the data archived?

Lines 340-341: Barrett et al. (2014) is not an appropriate citation for this tree-searching approach. This is a widely used method and does not require a citation.

Lines 448–458: I have two substantial concerns here. First, Arbour & Currie reported a minimum tree length of 420 steps, whereas your minimum tree length is 421. This suggests that you have not hit the minimum length trees in your analysis, despite using the New Technology search strategy. Second, New Technology searches tend to find a lower number of MPTs than ‘Traditional’ searches, and this is one reason why they should be followed by a round of TBR branch-swapping. The higher resolution in the strict consensus tree of your reanalysis of Arbour and Currie (2016) when using New Technology probably merely reflects the fact that you have not fully explored tree space and have not therefore collected all enough of the MPTs to adequately estimate the strict consensus. Your statement that ‘TBR branch-swapping on RAM trees was not possible due to computational limits, although this would not change the topology of the strict consensus’ is not strictly correct – adding this extra procedure will very likely reduce the resolution in the strict consensus tree, because you will explore more of the tree space and collect more MPTs. This extra procedure should therefore be implemented, and is unlikely to be hugely onerous in terms of computing memory requirements.

Line 520: Marginocephalia cannot be basal to Cerapoda, because Cerapoda is defined effectively as Marginocephalia + Ornithopoda. I’m not quite sure what you mean here.

Lines 630–631: How do we know that the topologies producing using supraspecific taxa by Butler et al. (2008) are correct?

Lines 750–770: This section is quite problematic for the reasons noted above (lines 448-458). You need to check that you are indeed hitting minimum length trees and that you are adequately exploring tree space.

·

Basic reporting

Excellent and detailed description of an important fossil. References up to date. Wonderful to see this important , but obscure taxon, refigured so well!

Lines 180 - 187 should be re-examined, I know I state, relative to Europelta, "such that the palate would not have had a pronounced hourglass appearance typical of derived nodosaurs such as Pawpawsaurus, Edmontonia, and Panoplosaurus [73–75]. "

I think the introductory part of the abstract could be tightened up a bit and the confirmation of another Lower Cretaceous stegosaurian occurrence on another continent given a bet more emphasis.

Experimental design

The phylogenetic analysis is very thorough. I particularly like their observation that higher taxonomic units used in a phylogenetic analysis need to be represented by both a basal and derived taxon.

Validity of the findings

In a descriptive paper such as this, exceptional figures are critical in evaluating the descriptions. Only 3D models of the described elements would have yielded more information.

On carefully reading the report, this reviewer is convinced as to the importance of this fossil material.

Additional comments

no additional comments

·

Basic reporting

See general comments.

Experimental design

See general comments.

Validity of the findings

See general comments.

Additional comments

Many thanks for the opportunity to review this manuscript – Paranthodon is a biogeographically significant dinosaur that is frustratingly poorly preserved, so any new insights into the evolutionary affinities of this taxon are sorely needed. I have a few comments that I hope will improve the manuscript, and also attach a marked-up PDF with other minor comments.

1. I think it would be helpful to introduce Isaberrysaura as another potential Gondwanan stegosaur – it jumped immediately to mind when I read “The geographic location of Paranthodon is particularly significant because it represents one of only two Gondwanan stegosaurs”. I know you discuss this when you get to your phylogenetic analyses, but I think others may notice its absence in this spot too, and it would be good to briefly mention it.

2. I appreciate the use of multiple analyses and comparisons to try to figure out just what Paranthodon really is, and I like the approach you’ve taken here. One thought I had while reading your discussion of exemplar taxa is that one of your main questions boils down to ‘is Paranthodon a stegosaur or an ankylosaur’ but you don’t have a dataset that will easily answer that question – you’ve got matrices that heavily sample ankylosaurs but lightly sample stegosaurs, and matrices that heavily sample stegosaurs but not ankylosaurs, but not an overall Thyreophora dataset. I know it’s not as simple as mashing together two character matrices and calling it a day, but perhaps there should be a coming together of Team Ankylosaur and Team Stegosaur to do just that and find out what some of these weirdo taxa are once and for all – sound like a plan for the future? In the meantime, I think there should be a comment somewhere in your methods section noting that there isn’t a comprehensive Thyreophora matrix at the moment and creating one is outside the scope of your project.

3. Relatedly, you might want to try sticking Paranthodon in the Arbour, Gates & Zanno 2016 dataset that included more comprehensive nodosaurid sampling in the ankylosaurid characters (and got pretty good resolution within Nodosauridae), especially as a comparison to the Thompson et al. matrix.

4. In the “Importance of Paranthodon” section, I think you need to be careful with the word ‘confirmed’. I think you can discuss which hypotheses of relationships for Paranthodon you can reject, and which are best supported, but since Paranthodon wanders around within Thyreophora a bit, I don’t think you can say you’ve confirmed it’s a stegosaur.

5. The photographs of Paranthodon are excellent but I think the figures could use a few more labels, or perhaps a more complete skull as reference just to orient readers who are less familiar with isolated thyreophoran skull bones.

6. You’ve got a figure summarizing the previously published phylogenies used in your analyses, but not one summarizing the results of your own analyses! I know you ran so many that it would be hard to fit all in one figure, but I think you could choose some exemplar results to summarize in one figure, with Paranthodon in bold, to help show how its position varied.

All in all a welcome addition to the thyreophoran literature, and I look forward to seeing the final version published!

---

## Round 0.2 · accepted · Accept

Thank you for your close attention to the comments from the reviewers. Everything seems to be in order at this point.